# Cost-effectiveness results comparing heat-stable carbetocin & other uterotonics in postpartum heamorrhage prevention in Uganda

Sam Ononge[1,2]*, Othman Kakaire[1], Jostas Mwembezi[3], Hadijah Nakatudde[4], Robert Mutumba[4], Richard Mugahi[4]

1 Department of Obstetrics and Gynaecology, Makerere University College of Health Sciences, Kampala, Uganda, 2 Department of Obstetrics and Gynaecology, Jinja Regional Referral Hospital, Jinja, Uganda, 3 Rwenzori Center for Research and Advocacy, Fort-Portal, Uganda, 4 Reproductive and Infant Health Division Ministry of Health, Kampala, Uganda

* ononge2006@yahoo.com

## Abstract

In Uganda, postpartum haemorrhage (PPH) is responsible for 34% of all institutional maternal deaths. Injectable oxytocin is the preferred uterotonic for prevention of PPH. However, in resource-limited settings, the effectiveness of oxytocin is sub-optimal due to efficacy, quality (cold chain storage requirements), and manufacturing standards (poor quality active pharmaceutical ingredients, lack of sterile manufacturing environment, and low-quality manufacturing processes). This study aimed to assess the cost-effectiveness of heat-stable carbetocin, a quality uterotonic newly recommended by WHO and Ugandan Ministry of Health, compared to standard uterotonics for the prevention of PPH in Uganda. A decision tree model was built to assess the cost-effectiveness of heat-stable carbetocin compared to the current standards of care – oxytocin, misoprostol or oxytocoin+misoprostol combination. The model was applied to a hypothetical annual cohort of birthing women eligible for PPH prevention in Uganda's public health facilities. The evaluation considered direct costs and health outcomes using a public health perspective. The model inputs were obtained through literature review and, whenever referencable information was unavailable or incomplete, from key opinion leaders. Compared to oxytocin, administering heat-stable carbetocin to prevent PPH had a cascading favorable effect and was estimated to avert 57,536 PPH cases, 123 maternal deaths, and 4,203 disability-adjusted life years (DALYs). Heat-stable carbetocin is also cost-saving where the direct cost to the public healthcare system was lower by USD $1,058,353 (UGX 3,998,350,875). The benefits of heat-stable carbetocin were even greater when compared with misoprostol (averted 73,939 PPH events, 273 maternal deaths, and 8,716 DALYs, and lowered public healthcare system costs by USD $2,118,372 [UGX 8,002,996,052]). Heat-stable carbetocin for preventing PPH in Uganda has the potential to reduce PPH events, and subsequently maternal deaths, DALYs, and costs for the public healthcare system. Adopting heat-stable carbetocin will contribute towards achieving the country's Sustainable Development Goal 3.1.

**Data availability statement:** All relevant data are within the paper and its Supporting Information files.

**Funding:** This study was financially supported by Ferring Pharmaceuticals via the Association of Obstetricians and Gynaecologists of Uganda in the form of an award. No additional external funding was received for this study. The funder had no role in study design, data collection and analysis, decision to publish, or preparation of the manuscript.

**Competing interests:** There are no patents, products in development or marketed products associated with this research to declare.

## Introduction

Investment in maternal health not only improves health outcomes, but also significantly benefits the overall socio-economic development of society [1,2]. While financing for maternal health, particularly for life-saving interventions, has seen steady improvement, several critical factors such as sustainable and efficient funding, relevant policies, and a reliable supply of quality commodities and services are still lacking [3]. By optimizing the utilization of scarce resources through the implementation of cost-effective interventions, we have the potential to prevent maternal morbidity and mortality. However, despite a 34% global reduction in maternal mortality between 2000 and 2020, more than a quarter of a million women died as a result of pregnancy or childbirth-related complications [4]. Postpartum haemorrhage (PPH), excessive bleeding from the birth canal in the first 24 hours after the birth of the baby, is traditionally defined as blood loss of >500 mL following vaginal or cesarean birth, and it is responsible, globally, for 20% of these maternal deaths [5]. PPH affects 14 million women each year, and 70,000 of these women die [6], particularly in low-income countries [7]. In Uganda, PPH is responsible for 34% of all institutional maternal deaths [8]. In addition to mortality, PPH is the leading cause of severe maternal morbidity, resulting in significant long-term and short-term consequences. These consequences include the acute effects of hemorrhagic shock and its management, such as multiorgan failure, transfusion-related morbidity, infection, chronic anaemia, intensive care admission, and prolonged hospitalization [9,10].

Uterine atony is the most common cause of PPH (70–80 percent) [11,12] and can be caused by anything that interferes with the ability of the uterus to contract and retract. Although uterine atony can occur in low-risk cases, interference with uterine contraction and retraction is most likely in the following clinical situations: labour induction, prolonged labour, precipitate labour, and uterine over-distension (e.g., multiple pregnancy, macrosomia, and polyhydramnios). In addition, uterine atony may occur in: retained blood clots (may result from a poorly contracting uterus following placental delivery), tocolytic drugs (glyceryl trinitrate or terbutaline, deep general anesthesia—particularly with fluorinated hydrocarbons), structural abnormalities of the uterus including uterine anomalies and fibroids, placenta previa (which limits the ability of the lower uterine segment to contract and retract), and preterm labour [13].

Management of PPH begins by prevention, and the World Health Organisation (WHO) considers uterotonics as the key component in active management of third stage of labour (AMTSL). Existing evidence shows that uterotonic drugs administered immediately after the birth of a baby reduce PPH by 66% [14]. In Uganda, the uterotonics used for prevention of PPH are oxytocin and misoprostol. Injectable oxytocin is the preferred drug of choice for prevention of PPH. However, in resource-limited settings, the sub-optimal effectiveness of oxytocin is due to efficacy [15] and quality issues, including inconsistent manufacturing standards (poor quality active pharmaceutical ingredients, lack of sterile manufacturing environment, and low-quality manufacturing processes) and fragile/dysfunctional cold-chain storage systems [16–23]. Maintenance of consistent cold-chain storage in low and middle-income countries (LMIC) is challenged by resources and infrastructural constraints, consequently, the quality of oxytocin in these settings is often below the international quality specifications [16–18]. Two other uterotonics that do not require refrigeration are misoprostol and heat-stable carbetocin. However, misoprostol is less effective than oxytocin in preventing PPH, is associated with more side effects and has persistent quality concerns when exposed to humidity [15,24]. Studies have consistently shown that the quality of oxytocin and misoprostol available in low-resource settings is often compromised [17,19–21]. Meanwhile, carbetocin is a long-acting uterotonic, safe, and effective in prevention of PPH. The heat-stable formulation of carbetocin can withstand extended exposure to high temperature and has a shelf life of 48

months at 30ºC [25]. According to the meta-analysis by Gallos et al, heat-stable carbetocin was associated with higher efficacy than oxytocin [15,26]. Based on the results of this meta-analysis, WHO revised its recommendations and essential medicines list to include heat-stable carbetocin for the prevention of PPH [26,27]. However, limited understanding among policymakers about the impact of integrating heat-stable carbetocin into care of pregnant women has delayed its introduction in many countries.

Uganda is a PPH high-burden setting [28], with more than three-quarters of women having childbirth at primary health facilities. The majority of these primary healthcare facilities have challenges in maintaining consistent cold-chain. In 2022, heat-stable carbetocin was added to Uganda's national management guidelines for PPH prevention [29] and essential medicines and health supplies list for Uganda 2023 [30]. This decision was made based on heat-stable carbetocin's proven heat-stability, and the aggregate of its efficacy and safety profile compared to other uterotonics used to prevent PPH [15], as well as the WHO Recommendation [27].

Few studies have compared the cost-effectiveness of heat-stable carbetocin with standard of care in LMIC [31]. A recent Indian modeling study found that administering heat-stable carbetocin to prevent PPH events was cost-effective to the public health sector compared with oxytocin and misoprostol [31]. To contribute to this literature, and to further support local decision-makers with relevant information, we needed to cost and compare the introduction of heat-stable carbetocin in a range of contexts, including in Uganda. The purpose of this study was to assess the cost-effectiveness and budget impact of heat-stable carbetocin compared to oxytocin, misoprostol, and a combination of the latter drugs for PPH prophylaxis among women who gave birth vaginally and through cesarean section in public healthcare facilities in Uganda.

## Methods

### Overview of the model

The same Excel-based decision tree model that Cook et al used in India [31] was used to model the introduction of heat-stable carbetocin (intervention) for PPH prophylaxis compared to the current standard of care in Uganda, aligned with the WHO 2018 recommendations (Fig 1). The cost-effectiveness of heat-stable carbetocin for PPH prophylaxis was compared to injectable oxytocin, oral misoprostol, and oxytocin+misoprostol combination, the most commonly used uterotonics in Uganda public healthcare facilities. The cost-effectiveness of heat-stable carbetocin for prevention of PPH was evaluated from the Ugandan public healthcare system perspective. The model considered a cohort of women who were in 3rd stage of labour and had given birth either vaginally or by caesarean section. Caesarean section delivery increases the risk and severity of PPH. The intervention and the standard of care were modeled based on the current service delivery structures and policy context, taking into account the current use of uterotonics at public facilities. The public healthcare system perspective which the study took included direct costs of drugs, administration, human resource utilization, cold-chain, and follow-up. Additional direct costs included were costs for blood transfusion, hospitalization (duration of hospital stay), and referrals. No costs were added for uterotonic adverse events and mortality. Discounting is not included in the base case, as all costs are incurred over a short period of time. The model considered the quality concerns for oxytocin, however, not for misoprostol.

The decision tree, as described in Fig 1, illustrates the various pathways that women may take from birth to discharge or death. At each section of the decision tree, certain inputs are applied. In this study, the model was applied to a cohort of women who gave birth in public

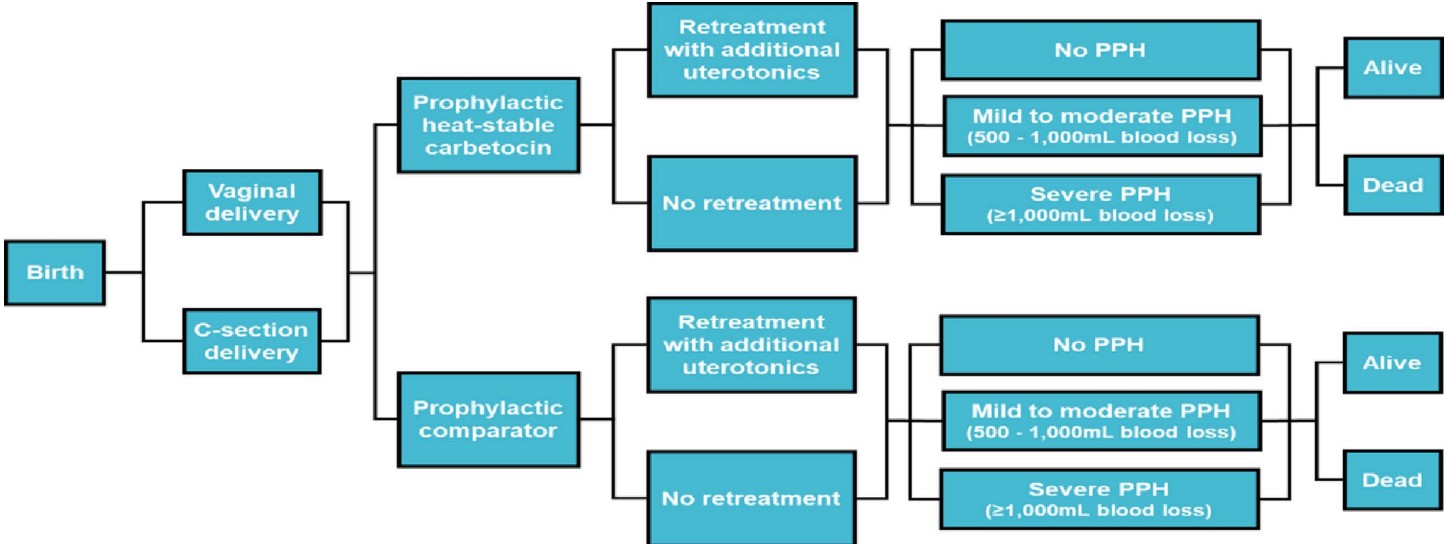

**Fig 1.  Model structure- Decision tree representing possible events in women receiving prophylactic heat-stable carbetocin versus comparator (oxytocin, misoprostol or oxytocin +misoprostol).** At each section of the decision tree, inputs are applied. These inputs are sourced via literature review or, where unavailable, key opinion leader guidance is captured. For simplicity, most non-pharmaceutical medical interventions were not included in this decision tree (e.g., UBT, NASG, hysterectomy, etc.) Key: C-section, caesarean section; Prophylactic comparator, oxytocin, misoprostol or oxytocin + misoprostol); Retreatment with additional uterotonics, oxytocin, misoprostol or oxytocin + misoprostol); mL, millilitre; PPH, postpartum haemorrhage. No PPH = blood loss of less than 500mL within 24 hours after birth. Mild-to-moderate PPH = blood loss of 500mL or more within 24 hours after birth. Severe PPH = blood loss of 1,000mL or more within 24 hours after birth.

healthcare facilities over a 1-year period. Even though women are given uterotonic medication to prevent PPH, some may still experience mild to moderate bleeding (500–1000 ml) or severe bleeding (more than 1000 ml) which would require additional medication for treatment. Women who suffer from severe PPH are at the greatest risk of death and usually need to be transferred to a higher-level health facility for appropriate care. The transfer costs are factored into the overall cost of treatment. The risk of death due to PPH and associated costs vary depending on whether the woman is being treated at a primary, secondary or tertiary healthcare facility. The decision tree is replicated for each setting in the model, and the results from each tree are weighted based on the proportion of deliveries occurring in each healthcare setting to obtain the overall results per prophylactic uterotonic chosen.

## Model inputs

The inputs for this decision-tree were obtained through a literature review, with a preference, when available, local Ugandan-produced literature. In cases where the information was not available or incomplete, input was sought from key opinion leaders. For the sake of simplicity, non-pharmaceutical medical interventions for PPH were not included in this decision-tree. Examples of such interventions excluded are uterine balloon tamponade, non-pneumonic anti-shock garments, B-lynch, and hysterectomy.

**Population and delivery characteristics.**  We considered a cohort of 1,172,523 women who gave birth at public health facilities in Uganda according to the annual health sector performance report of 2021/22 [32] (see Table 1 for health care setting and population characteristics). Among this group, approximately 25% were teenage mothers. Trained midwives assisted most of these childbirths. Out of the total number of women in the cohort, 76% delivered from primary facilities (Health Centre II and III), 16% from secondary facilities (Health Centre IV and district/general hospitals), and 8% from tertiary facilities (Regional and National Referral

**Table 1.** Health care setting and population characteristics.

| Input parameter | Estimate | Source of information |
|---|---|---|
| Annual deliveries | 1,172,523 | Annual Health Sector Performance report year 2021/22 [32] |
| Health facility where childbirth was done | | Annual Health Sector Performance report year 2021/22 [32] |
| Primary (HCII & HCIII) | 76% | |
| Secondary (HCIV & District hospital) | 16% | |
| Tertiary (Regional and National referral) | 8% | |
| Proportion delivered by C-Section | 10.5% | Annual Health Sector Performance report year 2021/22[32] |
| Distribution of women by age (years) | | National census report 2014 [35] |
| 15-19 | 24.8% | |
| 20-24 | 21.3% | |
| 25-29 | 16.7% | |
| 30-34 | 13.0% | |
| 35-39 | 10.1% | |
| 40-44 | 8.1% | |
| 45-49 | 5.9% | |
| Life expectancy for females (years) | 64.38 | UN World population projects 2022 [36] |
| Life expectancy discount rate for future years (n%) | 3% | Shanmugam KR 2011[37] |
| Proportion of pregnant women with anaemia | 30% | Bongomin et al 2021 [33] |
| Risk of PPH among women with anaemia (aOR) | 1.23 | WOMAN-2 trial, Lancet Glob Health 2023; 11: e1249–59 [34] |

Hospitals). 10.5% of women underwent caesarean section during delivery. It's worth noting that all childbirth services at public health facilities are provided by the government at no cost to the mother or her family. According to a systematic review by Bongomin et al, 30% of pregnant women in Uganda suffer from anaemia [33]. Considering this, we based the risk of a severe PPH event due to anaemia to data from WOMAN-2 trial [34]. The trial found that the odds of severe PPH of 1.23 in women with anaemia compared to those without.

**Efficacy of the prophylactic uterotonics (heat-stable carbetocin, oxytocin, misoprostol and oxytocin+misoprostol combination).** For this study, we relied on the efficacy of the prophylactic uterotonics based on the December 2018 Cochrane review on uterotonic agents for PPH prophylaxis [15]. This updated meta-analysis (U-MNA) included a total of 135,559 women from 196 trials conducted across all three socio-economic settings (high, middle and low-income status). As shown in Table 2, the primary outcomes in the meta-analysis were the relative effectiveness of each uterotonic for prevention of mild/moderate and severe PPH. A network meta-analysis (U-NMA) found that combinations of oxytocin with misoprostol or ergometrine, and stand-alone carbetocin, showed the highest efficacy. However, the combination regimens had more side effects, while carbetocin had none compared to oxytocin. The U-NMA concluded with "Carbetocin may be more effective than oxytocin for some outcomes without an increase in side effects" [15]. Tort et al provided information on the risk of PPH by level of facility and mode of delivery as well as the proportion of women that would require additional uterotonics and blood transfusion [38].

**Economic inputs.** The analysis included three categories of costs: uterotonic costs, management of PPH costs and post-treatment costs (Table 3). The prophylactic uterotonic costs considered in the model included drug acquisition costs for injectable oxytocin (10

**Table 2. Risk of PPH, use of additional uterotonics and blood transfusion based on Tort et al [38], Cochrane review updated Meta-analysis by Gallos [15] and duration of hospital stay [32].**

| Intervention | Risk of PPH | | | | | | Additional uterotonic | Blood transfusion |
|---|---|---|---|---|---|---|---|---|
| | Primary facility | | Secondary facility | | Tertiary facility | | | |
| | Mild/moderate PPH | Severe PPH | Mild/moderate PPH | Severe PPH | Mild/moderate PPH | Severe PPH | | |
| **Vaginal delivery** | | | | | | | | |
| Heat stable carbetocin | 6.4% | 2.87% | 6.25% | 2.73% | 6.1% | 2.6% | 5.20% | 1.2% |
| Oxytocin | 9.6% | 3.3% | 9.4% | 3.15% | 9.2% | 3.0% | 11.6% | 1.5% |
| Misoprostol | 10.0% | 3.96% | 9.8% | 3.78% | 9.6% | 3.6% | 12.1% | 1.3% |
| Oxytocin+ Misoprostol | | | | | 6.0% | 2.5% | 6.6% | 1.3% |
| **C-section delivery** | | | | | | | | |
| Heat stable carbetocin | 32.19% | 12.48% | 32.04% | 12.04% | 31.9% | 11.6% | 13.7% | 6.6% |
| Oxytocin | 47.13% | 14.26% | 47.12% | 13.78% | 47.1% | 13.3% | 30.4% | 8.1% |
| Misoprostol | 49.27% | 16.85% | 49.33% | 16.33% | 49.4% | 15.8% | 31.6% | 7.1% |
| Oxytocin+ Misoprostol | | | | | 29.9% | 12.4% | 17.3% | 6.9% |
| Duration of hospital stay (days) | 1.0 | 3.0 | 1.0 | 3.5 | 1.0 | 5.3 | | |

**Table 3. Parameters that influence economic cost used in the model. All costs in 2023 USD.**

| Parameter | Value | Source |
|---|---|---|
| Drug costs per dose (PPH prevention) | | |
| Injectable Oxytocin (10 IU) | USD 0.15 | National Medical Stores (Uganda), 2023 |
| Misoprostol (600 µg) | USD 0.38 | National Medical Stores (Uganda), 2023 |
| Heat stable carbetocin (100 µg) | USD 0.62 | Ferring Pharmaceuticals (for medicine's ex-man price) + transportation/clearance + National Medical Stores PSM fees |
| Oxytocin dosing distribution for PPH prevention | | |
| 5 IU | 11.3% | Ejekam, et al 2019 [39] |
| 10 IU | 32.6% | |
| 15 IU | 4.8% | |
| 20 IU | 41.4% | |
| 30-60 IU | 9.9% | |
| Misoprostol dose for PPH prophylaxis (600 µg) | 100% | Essential Maternal and Newborn Clinical Care Guidelines for Uganda (2022) |
| Proportion of women who receive both oxytocin and misoprostol for PPH prevention by setting | | Key Opinion Leader |
| Primary health centre | 30% | |
| Secondary health facility | 40% | |
| Tertiary health facility | 70% | |
| Proportion of women who receive oxytocin and misoprostol for treatment of PPH | 100% | Key Opinion Leader |
| Dose of oxytocin for treatment of PPH | 20 IU | Essential Maternal and Newborn Clinical Care Guidelines for Uganda (2022) |
| Dose of misoprostol for treatment of PPH | 800 µg | |
| Cold chain cost for an ampoule of oxytocin | | |
| During transport, testing & management per ampoule | USD 0.006 | National Medical Stores (Uganda), 2023 |
| Cost for storage at the facility per ampoule | USD 0.045 | Access Health International 2018 |
| % of ampoules disposed off | 32% | Vlassoff et al 2016 [40] Diop et al 2016 [41] |
| Cost of syringe for IV/IM injection | USD 0.039 | National Medical Stores (Uganda), 2023 |

*(Continued)*

**Table 3.** (Continued)

| Parameter | Value | Source |
|---|---|---|
| Time taken per doctor to manage PPH in vaginal delivery (in minutes) | | Key Opinion Leader |
| No PPH | 30 | |
| Mild/moderate/severe | 60 | |
| Time taken per a nurse to manage PPH in Vaginal delivery (in minutes) | | |
| No PPH | 30 | |
| Mild/moderate PPH | 60 | |
| Severe PPH | 90 | |
| Time taken per auxillary nurse to manage PPH in a vaginal delivery (in minutes) | | |
| No PPH | 30 | |
| Mild/moderate PPH | 60 | |
| Severe PPH | 90 | |
| Time taken per doctor to manage PPH in caesarean section delivery (in minutes) | | |
| No PPH | 60 | |
| Mild/moderate PPH | 60 | |
| Severe PPH | 90 | |
| Time taken per a nurse to manage PPH in caesarean section delivery (in minutes) | | |
| No PPH | 30 | |
| Mild/moderate PP | 60 | |
| Severe PPH | 90 | |
| Time taken per auxillary nurse to manage PPH in caesarean section delivery (in minutes) | | |
| No PPH | 30 | |
| Mild/moderate PPH | 60 | |
| Severe PPH | 90 | |
| Amount of blood units used in | | |
| No PPH | 0 | Key Opinion Leader |
| Mild/moderate PPH | 1 | |
| Severe PPH | 3 | |
| Salary of the health providers (monthly) | | |
| Doctor/physician | USD 1,431.93 | Public service Government salary structure (year 22/23) |
| Nurse | USD 376.38 | |
| Auxiliary nurse | USD 157.79 | |

IU) and misoprostol (200 μg) as per the National Medical Stores (NMS, the Ugandan government's purchasing authority). Meanwhile the cost for heat-stable carbetocin (100 μg) was provided by Ferring Pharmaceuticals, adding transport and NMS inputs. The national management guidelines recommend using 10 international units of oxytocin for PPH prophylaxis. However, issues regarding the quality of oxytocin influence how physicians prescribe this medication. We utilized data from a Nigerian study by Ejekam et al [39] to estimate this behaviour. In addition, we consulted with key opinion leaders about the quality issues of oxytocin. They provided us with the information that, due to oxytocin quality, some birthing women were likely to receive oxytocin+misoprostol combination for prophylaxis and

this was higher in tertiary facilities than in secondary/primary facilities. This is due to higher rates of caesarean section in tertiary facilities. The national clinical guidelines provided the information on the dose of both oxytocin and misoprostol for birthing woman experiencing PPH. Meanwhile, the NMS provided the estimates of the cold-chain costs associated with transporting oxytocin, testing as well as the cost of syringes. The NMS estimates 8.5% of invoice price for transport and cold-chain till the facility. We did not have a local cost estimate of cold chain storage at health facilities. However, we obtained input from a study from India.

The cost of wastage was taken into account to cover the cost of drugs that are unusable due to heat, expiration date, breakage, and wear and tear. In the case of injectable oxytocin, the estimated wastage rate was 32%. This estimate is based on two separate studies conducted in Senegal one by Vlassoff et al. [40] and the other by Diop et al using oxytocin Uniject [41].

Based on the data collected from Key Opinion Leaders, the cost for hospital stay related with PPH at primary health facility was USD 16.51 per day while the same cost at secondary and tertiary facilities was USD 105.41. These costs apply to all patients, regardless of the severity of PPH, and include additional uterotonics, consumables, medical/non-medical cost, monitoring instruments and their maintenance. The cost of a single blood transfusion is USD 80, which includes the cost of blood acquisition, screening and administration. The estimated cost of referring a PPH patient from a primary to a secondary health facility is USD 26.4, while the estimated cost of referring a patient from a secondary to a tertiary health facility is USD 37.1.

Follow-up costs for PPH patients in Uganda often include a five-day course of antibiotics and 30 days of haematinics, as well as wound dressing for secondary infections and re-admissions, with an estimated cost of USD 1.59.

**Disability-adjusted life years.** The metric of disability-adjusted years is used to measure the burden of disease by taking into account both premature death and the time spent living with a disability. To calculate these metrics, we first determined the years of life lost due to premature death (in cases that result in fatality) and then computed the years lived with disability (in cases where the patient survives or experiences a non-fatal outcome). The years lost due to PPH mortality were accumulated from the loss of life due to premature death. To estimate the remaining life years lost for each death, life table information for females in Uganda was used. As DALYs due to fatal PPH events accumulate in the future, a discount rate of 3% per year was applied to the life years lost. Secondly, we determined the number of years that the survivors of PPH live with disability based on the severity of PPH and duration of disability caused by it. The severity of PPH was categorized as mild/moderate or severe, and the inputs were based on published data from Lubinga et al, Seligman and WHO expert opinion [42–44]. The average duration of disability due to mild/moderate PPH and severe PPH are assumed to be 30 days and 90 days, respectively. Disability weights for mild/moderate and severe PPH were drawn from the global burden of disease study published by Salomon et al [45]. The weight for severe PPH was 0.473 for days 0–30, 0.324 for days 31–90. The weight for mild/moderate was 0.166.

**Risk of mortality due to PPH.** We do not have direct estimates for the risk of mortality due to PPH in Uganda. Instead, we used the maternal mortality ratio (MMR) and the percentage of deaths due to PPH to calculate the expected number of deaths due to PPH per 100,000 women. We then divided this number by the estimated number of PPH events per 100,000 women, which is based on the efficacy of each prophylactic uterotonic and their respective market share. The MMR for Uganda is 189 (according to Uganda Demographic Health Survey 2022) [46], with 34% of the deaths being due to PPH (Annual maternal death review report 2022/23) [47]. Assuming a current market share of prophylactic uterotonics of 61% oxytocin, 5% misoprostol, and 34% oxytocin+misoprostol combination, the overall risk of mortality due to PPH is 0.41%.

The model required an estimation of the risk of death caused by PPH, which is stratified by the severity of PPH and the type of healthcare centre. The study conducted by Tort et al in 2015, provided odds ratios for mortality based on several factors including maternal age, type of healthcare facility, actual referrals to another facility, and severity of PPH. The risk of mortality was 2.43 for primary health facility, 1.49 for secondary health facility and 1.00 for tertiary healthcare facility, and maternal age (1.00 for ages less than 20 years; 1.48 for ages between 20 and 35 years; 2.16 for ages above 35 years) [33]. The resulting estimates for the risk of mortality due to PPH, stratified by age, healthcare setting and severity of PPH, are shown in Table 4. The odds ratio for mortality during referral to another hospital is estimated to be 13.35. According to key opinion leaders, not all women experiencing a PPH event while in a primary healthcare facility should be referred to a secondary or tertiary healthcare facility if the bleeding is under control. Only 80% of them are actually transferred from primary to secondary/tertiary healthcare facilities, while only 4% are referred from a secondary health facility to a tertiary facility due to the availability of comprehensive obstetric services at secondary facilities.

**Budget impact analysis.** A budget impact analysis was also conducted by projecting the annual cost to the healthcare system for births at public institutions based on the current estimated market share of prophylactic uterotonics ("current environment") and based on a potential shift in market share with the entry and uptake of heat-stable carbetocin ("new environment") over the next 5 years (Table 5). The analysis also considers the total number of births from 2024 through 2028, which was estimated using a crude rate of 37 births per 1,000 persons in a population of 47,249,585 in 2022 [48] with an annual growth rate of 3.0%. Of these births, 91% are institutional births, of which 80.5% occur in a public facility. The

Table 4. Inputs of mortality data following PPH event stratified by age, healthcare setting and severity of PPH.

| Input parameter | Value | Source |
|---|---|---|
| Risk of mortality due to PPH by age group (OR) | | Tort et al (2015) [38] |
| Less than 20 years | 1 | |
| Between 20 and 35 years | 1.48 | |
| More than 35 years | 2.16 | |
| Risk of PPH-related mortality by location of healthcare setting | | Tort et al 2015 [38] |
| Primary health facility | 1.69 | |
| Secondary health facility | 1.30 | |
| Tertiary health facility | 1 | |
| Proportion of PPH-related referrals that actually get transferred (%) | | Clinical and Key Opinion Leader |
| From primary to secondary health facility | 80% | |
| From secondary to Tertiary health facility | 4% | |
| Odds ratio of mortality due to referral to another healthcare setting for PPH | 13.35 | Tort et al 2015 [38] |
| Overall risk of death after a PPH event (%) | 0.41% | |
| Maternal mortality ratio per 100,000 live births (MMR) | 189 | Uganda Demographic Health Survey 2022 |
| PPH-related mortality as percentage of the MMR above (%) | 34.0% | Annual MPDSR report 2022/23 |

**Table 5. Public facility-based uterotonic market share – prophylactic dosing only.**

| Current scenario (without heat-stable carbetocin) | | | | | |
|---|---|---|---|---|---|
| Treatment | 2024 | 2025 | 2026 | 2027 | 2028 |
| Oxytocin | 61.00% | 61.00% | 61.00% | 61.00% | 61.00% |
| Misoprostol | 5.00% | 5.00% | 5.00% | 5.00% | 5.00% |
| Oxytocin+Misoprostol combination | 34.00% | 34.00% | 34.00% | 34.00% | 34.00% |
| New scenario (with heat-stable carbetocin) – prophylactic dosing only | | | | | |
| Heat-stable carbetocin | 2.00% | 5.00% | 10.00% | 20.00% | 30.00% |
| Oxytocin | 60.00% | 58.00% | 55.00% | 50.00% | 45.00% |
| Misoprostol | 5.00% | 5.00% | 5.00% | 5.00% | 5.00% |
| Oxytocin+Misoprostol combination | 33.00% | 32.00% | 30.00% | 25.00% | 20.00% |

distribution of births by healthcare facility type and mode of delivery, along with clinical and healthcare resource use inputs, are the same as used in the cost-effectiveness analysis.

## Results

### Cost-effectiveness analysis

In comparison with other uterotonics related to health outcomes, the use of heat-stable carbetocin for preventing postpartum haemorrhage (PPH) resulted in fewer all PPH events, severe PPH cases, deaths and disability-adjusted life years (DALYs) (see Table 6). When compared individually to oxytocin, misoprostol, and oxytocin+misoprostol combination, the use of heat-stable carbetocin for PPH prevention resulted in fewer cases of PPH by 28.4%, 33.8%, and 20.0%, respectively. A similar trend was observed with the number of severe PPH cases (by 13.1%, 27.4% and 8.9% respectively), deaths (by 15.0%, 28.1% and 10.8%, respectively), and DALYs (by 16.1%, 28.5% and 11.4%, respectively). Total costs to the public healthcare system are also lowest when heat-stable carbetocin is selected as the prophylactic uterotonic administered (see Table 6). When compared individually to oxytocin, misoprostol, and oxytocin+misoprostol combination, the use of heat-stable carbetocin for PPH prevention resulted in lower costs by 1.7%, 3.5% and 1.0%, respectively.

Based on this analysis, where heat-stable carbetocin out-performed oxytocin, misoprostol, and oxytocin+misoprostol combination in terms of health outcomes and healthcare costs, heat-stable carbetocin was cost-effective from the Uganda public health care system perspective.

Table 7 provides a breakdown of individual costs of the public healthcare system related to PPH management (prevention and treatment). The lower total cost when heat-stable carbetocin was administered to prevent PPH can be mostly associated with: lower hospital stay costs (savings of USD 334,854 – 1,621,592 or UGX 1,265,045,793 – 6,126,211,011); lower blood

**Table 6. Base case analysis for the cohort of 1,172,523 women giving birth at public facilities.**

| Intervention | All PPH Events | Severe PPH Events | Deaths | DALYs | Total costs (USD)* | Total costs (UGX) |
|---|---|---|---|---|---|---|
| Heat-stable carbetocin | 144,853 | 41,566 | 699 | 21,832 | 60,569,720 | 228,826,345,770 |
| Oxytocin | 202,389 | 47,857 | 822 | 26,035 | 61,628,073 | 232,824,696,646 |
| Misoprostol | 218,793 | 57,231 | 972 | 30,548 | 62,688,092 | 236,829,341,822 |
| Oxytocin+Misoprostol@ | 181,122 | 45,645 | 783 | 24,650 | 61,170,762 | 231,097,020,381 |

* Total costs in USD were converted from the UGX Total costs column using the following exchange rate - 3,777.90 UGX to 1 USD (Accessed 29 Nov. 2023 - Currency Converter | Foreign Exchange Rates | OANDA)*

Note: all local (UGX) economic-related parameters represented in USD in this manuscript used this exchange rate for conversion.

**Table 7. Detailed distribution of costs to the public healthcare system for the cohort of *1,172,523 women (in USD)*.**

| All births, whole cohort | Heat-stable carbetocin | Oxytocin | Misoprostol | Oxytocin+Misoprostol@ |
|---|---|---|---|---|
| **Total Costs (USD)** | **60,569,720** | **61,628,073** | **62,688,092** | **61,170,762** |
| Hospital stay costs | 48,676,380 | 49,327,565 | 50,297,972 | 49,011,234 |
| Healthcare personnel costs | 7,897,608 | 8,114,438 | 8,187,609 | 8,033,633 |
| Follow-up costs | 1,899,426 | 1,899,426 | 1,899,426 | 1,899,426 |
| Drug costs | 781,613 | 439,967 | 584,964 | 576,863 |
| Referral costs | 678,804 | 781,534 | 934,624 | 750,248 |
| Blood transfusion costs | 419,184 | 704,560 | 671,895 | 525,665 |
| Administration costs | 209,625 | 228,590 | 95,161 | 244,065 |
| Cold-chain costs | 7081 | 131,994 | 16,442 | 129,628 |
| *Cold-chain costs: logistics* | 406 | 7,579 | 944 | 7,443 |
| *Cold-chain costs: storage* | 3,177 | 59,219 | 7,377 | 58,157 |
| *Cold-chain costs: wastage* | 3,497 | 65,197 | 8,121 | 64,028 |

transfusion costs (savings of USD 106,480 – 285,375 or UGX 402,272,507 – 1,078,119,587) and lower health personnel costs (savings of USD 136,026 – 290,002 or UGX 513,891,009 – 1,095,596,762). These lower costs were attributed to decreased severe PPH events and fewer days of inpatient stay. Drug costs were the only cost category where heat-stable carbetocin was invariably higher compared to the other uterotonics. The largest cost savings from utilizing heat-stable carbetocin for prevention of PPH compared to the other uterotonics were associated with lower hospital stay costs, as per above, attributed to the decrease in severe PPH events and fewer days of inpatient stay.

## Probability sensitivity analysis

One-way sensitivity analysis (OWSA) was performed to identify the key drivers of the cost-effectiveness of heat-stable carbetocin versus oxytocin. The analysis considered individual model parameters associated with uncertainty and structural assumptions or settings that have a notable impact on the estimates of costs and DALYs for each intervention. The upper and lower bound estimates for parameters were taken from the reported 95% confidence intervals (CIs) and reported ranges. When there were no CIs available, we assumed a standard error of 20%. Because heat-stable carbetocin dominated oxytocin, the OWSA was conducted separately for the incremental DALYs and incremental costs. The result of the probability sensitivity analysis (PSA) is displayed in Fig 2, (showing the Probability sensitivity analysis results for incremental costs and DALYs averted: Heat-stable carbetocin versus oxytocin), with the PSA replicates of the incremental costs and DALYs avoided for heat-stable carbetocin versus oxytocin. Heat-stable carbetocin out-performs oxytocin. Over 95% of the PSA replicates indicate that heat-stable carbetocin lowers costs and averts more DALYs than oxytocin.

## Budget impact analysis

From 2024 through 2028 approximately 7,069,055 cumulative births were projected in public healthcare facilities in Uganda. During this period, costs to the public healthcare system are projected to decrease by USD 651,636 or UGX 2,461,813,868 based on the incremental growth in heat-stable carbetocin integration reaching 30% of prophylactic uterotonic use by 2028. The budget savings are driven by approximately 25,060 fewer PPH events projected to occur. See Table 8 for the projected annual budget and clinical impact of introducing heat-stable carbetocin over the 2024–2028 period in the public sector of Uganda.

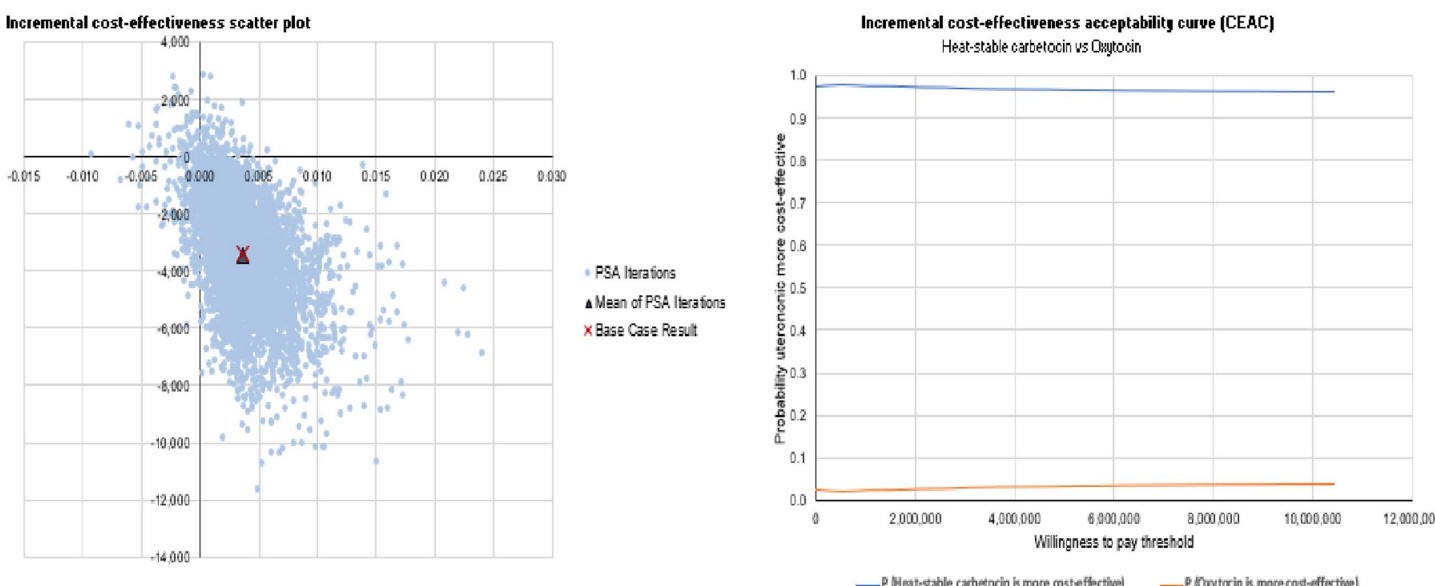

| X-Axis: DALYs Averted |
| :---: |
| Y-Axis: Incremental Costs |

| CE Scatterplot Quadrant | Heat-stable carbetocin vs. Oxytocin | | Probability |
| --- | --- | --- | --- |
| | Incremental Cost | DALYs Averted | |
| SE | < 0 | ≥ 0 | 0.9510 |
| NE | ≥ 0 | ≥ 0 | 0.0154 |
| SW | < 0 | < 0 | 0.0222 |
| NW | ≥ 0 | < 0 | 0.0114 |

**Fig 2. Probabilistic Sensitivity Analysis results for incremental costs and DALYs averted, heat-stable carbetocin versus oxytocin.** Heat-stable carbetocin has a >95% probability to save costs & avert more DALYs compared to oxytocin.

**Table 8. Projected annual budget and clinical impact of new versus current market share for uterotonics in Uganda.**

| | 2024 | 2025 | 2026 | 2027 | 2028 |
| --- | --- | --- | --- | --- | --- |
| **Total eligible births** | 1,331,489 | 1,371,433 | 1,412,576 | 1,454,954 | 1,498,602 |
| **Total costs (USD)** | | | | | |
| Current environment | 69,917,343 | 72,014,864 | 74,175,309 | 76,400,569 | 78,692,586 |
| New environment | 69,900,336 | 71,967,318 | 74,077,365 | 76,206,763 | 78,397,255 |
| Budget impact | (17,088) | (47,546) | (97,945) | (193,806) | (295,331) |
| **Total PPH events** | | | | | |
| Current environment | 206,406 | 212,598 | 218,976 | 225,545 | 232,312 |
| New environment | 205,795 | 210,667 | 214,998 | 218,107 | 221,210 |
| Clinical impact | (611) | (1,931) | (3,978) | (7,438) | (11,102) |

As shown in the Table 8, the increasing cost savings over time due to annual increases in the cohort of patients administered heat-stable carbetocin can be re-directed to other uses. This trend suggests that with increased uptake of heat-stable carbetocin, the cost savings would be even greater than projected here.

## Discussion

Preventing PPH is the first line of defense to protect women and reduce the burden on the healthcare system. Evidence shows that when quality-assured uterotonics are administered appropriately and in a timely fashion, AMTSL is highly effective at preventing PPH [14]. Efficacy differences among uterotonics and the use of low-quality uterotonics, however, contribute to today's sub-optimal outcomes.

The present cost-effectiveness analysis, framed within the Ugandan context, examined the direct economic costs and health outcomes of heat-stable carbetocin when administered for PPH prevention compared to standard prophylactic uterotonics and across the post-partum period from prevention through treatment of PPH (see Fig 1 Model structure- decision tree), with the intent to provide decision-makers with informed analysis that supports optimal health outcomes.

This analysis revealed that heat-stable carbetocin to prevent PPH provided better health outcomes and lower healthcare costs compared to oxytocin, misoprostol and oxytocin+misoprostol combination from the Ugandan public health care system perspective.

In individual comparisons, we observed that using heat-stable carbetocin instead of oxytocin resulted in a 28.4% decrease in PPH events. When compared to misoprostol, administering heat-stable carbetocin resulted in an even greater reduction in PPH events (33.8%). And when compared to the combination of oxytocin+misoprostol, heat-stable carbetocin resulted in fewer PPH events (20.0%).

Interestingly, this analysis also shows that despite the initial cost of purchasing heat-stable carbetocin being higher than recommended dosing of comparator prophylactic uterotonics, it was associated with better health outcomes and lower total costs related to PPH. These findings are consistent with other recent cost-effectiveness studies that compared carbetocin to alternative uterotonics: Cook et al. (2023) found that in India, heat-stable carbetocin resulted in cost savings compared to oxytocin and misoprostol due to the avoidance of PPH-related complications [31]; Pickering et al (2018) found that in the UK, carbetocin lowered the risk of total and severe PPH events, resulting in lower total costs compared to oxytocin [49]; and in Canada, Barrett et al (2022) found that cost savings attributed to carbetocin versus oxytocin, were primarily attributed to avoiding consequences of PPH [50].

In this analysis, across the entire cohort, while the difference in costs in percentage terms was limited (1.0% - 3.5%), in absolute terms (see Table 7), the increasing cost savings over time due to annual increases in the cohort of patients administered heat-stable carbetocin can be re-directed to other uses (see Tables 5 and 8). This trend suggests that with an accelerated uptake of heat-stable carbetocin, the cost savings could be even greater than projected here.

The use of heat-stable carbetocin for preventing PPH in low-resource settings, such as Uganda, can significantly improve the quality of care and promote health equity. This new heat-stable molecule can be particularly beneficial in places where electricity is unreliable, cold-chain transport and storage are challenging to maintain and high humidity is the norm. The problem with poor quality oxytocin and misoprostol used in LMICs to manage obstetric haemorrhage, consistently found in country studies and systemtic review [50], could be a contributing factor to the persistence of maternal deaths due to PPH in these settings, despite affordable and broad access.

These findings have important implications for Uganda, where PPH is a significant problem. It provides an opportunity to further optimize the country's efforts to reduce PPH, and thus achieve better outcomes. Heat-stable carbetocin, a new alternative that could, if fully integrated into policy and practice, facilitates a crucial step forward in supporting the government's priorities in women's healthcare, including contributing towards meeting the country's Sustainable Development Goal (SDG) 3.1 commitment to reduce maternal mortality.

Decision-makers, budget holders, and health administrators, increasingly asked to be efficient stewards of limited resources, should be interested to learn that the use of heat-stable carbetocin, administered prophylactically, resulted in the lowest total costs and the best health outcomes to the Ugandan public healthcare system, regardless of comparator prophylactic uterotonic, based on this analysis.

Introducing heat-stable carbetocin to prevent PPH can be a game-changer in the fight against PPH and maternal mortality by optimizing PPH-related health outcomes. Its cost-effectiveness, better health outcomes and lower costs, mark a crucial step forward in supporting the government's priorities in women's healthcare, including contributing towards meeting the country's SDG 3.1 commitment. Heat-stable carbetocin should be included in Uganda's PPH programming and essential health packages as a novel alternative that can support achievement of the country's sustainable development goals, specifically, the SDG on maternal mortality (3.1) and universal health coverage.

## Limitations to the study

There are some important limitations to consider in our modeling exercise that require further discussion. Firstly, our decision analytic model did not take into account the indirect costs to the healthcare system, such as healthcare provider training. Secondly, non-pharmaceutical medical interventions, including uterine balloon tamponade, non-pneumotic anti-shock garment, B-lynch, and hysterectomy, were excluded from the model, which underestimated the total economic costs of managing PPH events. In addition, ergometrine, which is an alternative uterotonic, was also excluded from the model due to its limited use for PPH prevention. The model also did not include tranexamic acid as an adjunct therapy for PPH treatment, which suggests an underestimation of the total economic costs and overall health outcomes. Furthermore, poor-quality misoprostol was excluded from the model, which suggests an underestimation of the total economic costs and health outcomes of its real-world administration, including additional dosing and PPH events, as well as wastage. Lastly, the cold-chain costs included in the model may be an underestimation of true costs, as studies have shown that cold-chain is not always available or maintained in LMICs or different LMIC settings.

## Supporting information

**S1 Data**   Data inputs into the model. PHC: Primary Health Care, SHC: Secondary Health Care, THC: Tertiary Health Care, PPH: Postpartum Hemorrhage, U-NMA: Uterotonic Network Meta-Analaysis, C-section: Cesarean section, KOL: Key Opinion Leaders, EMNCG: Essential maternal newborn Care guidelines.
(XLSX)

## Acknowledgments

The activities in this study were supported by MSD, through its MSD for Mothers initiative, an initiative of Merck & Co., Inc., Rahway, NJ, U.S.A. The authors are grateful to J. Cook et al. from the Indian study [31] that allowed the Excel-based model to be applied in a Ugandan

setting. We thank several Ugandan Key Opinion Leaders who gave their expert opinion and time to participate in the study.

## Author contributions

**Conceptualization:** Sam Ononge, Othman Kakaire, Jostas Mwembezi, Hadijah Nakatudde, Robert Mutumba, Richard Mugahi.

**Data curation:** Sam Ononge, Othman Kakaire, Jostas Mwembezi, Robert Mutumba.

**Funding acquisition:** Sam Ononge.

**Project administration:** Jostas Mwembezi, Hadijah Nakatudde.

**Supervision:** Othman Kakaire, Richard Mugahi.

**Validation:** Robert Mutumba.

**Writing – original draft:** Sam Ononge.

**Writing – review & editing:** Sam Ononge, Othman Kakaire, Jostas Mwembezi, Hadijah Nakatudde, Robert Mutumba, Richard Mugahi.

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
