## [Decision Letter · Decision Letter 0]

3 Dec 2024

PGPH-D-24-01624

Cost-effectiveness results comparing heat-stable carbetocin & other uterotonics in postpartum heamorrhage prevention in Uganda

Dear Dr. Ononge,

Thank you for submitting your manuscript to PLOS Global Public Health. After careful consideration, we feel that it has merit but does not fully meet PLOS Global Public Health’s publication criteria as it currently stands. Therefore, we invite you to submit a revised version of the manuscript that addresses the points raised during the review process.

Reviewer One's comments seem incomplete, but I have not been able to get additional feedback from them regarding whether they were somehow lost in transmission.  Please respond to them as best you can.

Please make sure to respond comprehensively to Reviewer Two's request for further justification of key model parameters.

We look forward to receiving your revised manuscript.

Kind regards,

Abraham D. Flaxman, Ph.D.

Academic Editor

Journal Requirements:

Additional Editor Comments (if provided):

Reviewers' comments:

Reviewer's Responses to Questions

**Comments to the Author**

1. Does this manuscript meet PLOS Global Public Health’s publication criteria?

Reviewer #1: Yes

Reviewer #2: No

2. Has the statistical analysis been performed appropriately and rigorously?

Reviewer #1: Yes

Reviewer #2: Yes

3. Have the authors made all data underlying the findings in their manuscript fully available (please refer to the Data Availability Statement at the start of the manuscript PDF file)?

Reviewer #1: Yes

Reviewer #2: Yes

4. Is the manuscript presented in an intelligible fashion and written in standard English?

Reviewer #1: Yes

Reviewer #2: Yes

Reviewer #1: This is a we written article and of global importance and makes a significant contribution to the management of PPH especially in low income countries. I have a few comments:

Review the sentence that starts with however on line number200, it sounds incomplete.

Line numbers 337 and 338 "When there was no CIs available, we assumed a standard error of 20%" What is the basis of spoor choice>

Reviewer #2: This is a valuable contribution and will provide important evidence for preventing post-partum hemorrhage. The authors must elaborate their justification of key model parameters, however.

Major issues:

I was not aware that HSC was proven superior to Oxy, as is implied by the first two rows of Table 2. I thought the RCT evidence showed “non-inferiority”.[1] In light of this, I think a much more extensive justification of Table 2 is required.

Line 257-259: what is the basis for the utilization rates you have assumed for Oxy, Miso, and combination?

Line 259: I was not able to replicate the finding that overall risk of mortality due to PPH is 0.41%. Please elaborate on how you have derived this key quantity.

I did not find the probability of hemorrhage when patients received both Oxy and Miso. Did I miss it? Consider adding to Table 2.

[1] Widmer M, Piaggio G, Abdel‐Aleem H, Carroli G, Chong Y, Coomarasamy A, et al. Room temperature stable carbetocin for the prevention of postpartum haemorrhage during the third stage of labour in women delivering vaginally: study protocol for a randomized controlled trial. Trials 2016;17:143.

**Do you want your identity to be public for this peer review?** For information about this choice, including consent withdrawal, please see our Privacy Policy

Reviewer #1: **Yes: ** Abigail Kazembe, PhD.

Reviewer #2: No

---

## [Decision Letter · Decision Letter 1]

18 Feb 2025

Cost-effectiveness results comparing heat-stable carbetocin & other uterotonics in postpartum heamorrhage prevention in Uganda

PGPH-D-24-01624R1

Dear Dr Ononge,

We are pleased to inform you that your manuscript 'Cost-effectiveness results comparing heat-stable carbetocin & other uterotonics in postpartum heamorrhage prevention in Uganda' has been provisionally accepted for publication in PLOS Global Public Health.

Best regards,

Abraham D. Flaxman, Ph.D.

Academic Editor

Reviewer Comments (if any, and for reference):

Reviewer's Responses to Questions

**Comments to the Author**

Reviewer #1: All comments have been addressed

publication criteria?

Reviewer #1: Yes

3. Has the statistical analysis been performed appropriately and rigorously?

Reviewer #1: Yes

4. Have the authors made all data underlying the findings in their manuscript fully available (please refer to the Data Availability Statement at the start of the manuscript PDF file)?

Reviewer #1: Yes

5. Is the manuscript presented in an intelligible fashion and written in standard English?

Reviewer #1: Yes

Reviewer #1: All comments that were raised have been addressed adequately

**Do you want your identity to be public for this peer review?** For information about this choice, including consent withdrawal, please see our Privacy Policy

Reviewer #1: No
